# Invasive *Candida parapsilosis* Bloodstream Infections in Children: The Antifungal Susceptibility, Clinical Characteristics and Impacts on Outcomes

**DOI:** 10.3390/microorganisms11051149

**Published:** 2023-04-28

**Authors:** Yao-Sheng Wang, Jen-Fu Hsu, Wei-Ju Lee, Shao-Hung Wang, Shih-Ming Chu, Hsuan-Rong Huang, Peng-Hong Yang, Ren-Huei Fu, Ming-Horng Tsai

**Affiliations:** 1College of Medicine, Chang Gung University, Taoyuan 244, Taiwan; 2Division of Pediatric Emergency Medicine, Department of Pediatrics, Chang Gung Memorial Hospital, Chiayi 613, Taiwan; 3Division of Pediatric Neonatology, Department of Pediatrics, Chang Gung Memorial Hospital, Taoyuan 244, Taiwan; 4Department of Microbiology Immunology and Biopharmaceuticals, National Chiayi University, Chiayi 613, Taiwan; 5Division of Neonatology and Pediatric Hematology/Oncology, Department of Pediatrics, Chang Gung Memorial Hospital, Yunlin 638, Taiwan

**Keywords:** candidemia, intensive care unit, *Candida parapsilosis*, bloodstream infection, mortality

## Abstract

**Background**: *Candida parapsilosis* is the most common non-*albicans* candida species that causes invasive candidiasis, but little is known about its impacts on the outcomes of pediatric patients. We aimed to characterize the clinical characteristics, risk factors and outcomes of *C. parapsilosis* bloodstream infections (BSIs) in children. **Methods**: All pediatric patients with *Candida parapsilosis* BSIs between 2005 and 2020 from a medical center in Taiwan were enrolled and analyzed. The antifungal susceptibility, clinical manifestations, management and outcomes were investigated. Cases of *Candida parapsilosis* BSIs were compared between patients with *C. albicans* BSIs and other Candida spp. BSIs. **Results**: During the study period, 95 episodes (26.0% of total cases) of *Candida parapsilosis* BSIs were identified and analyzed. No significant difference was found between pediatric patients with *C. parapsilosis* BSIs and those with *C. albicans* BSIs in terms of patients’ demographics, most chronic comorbidities or risk factors. Pediatric patients with *C. parapsilosis* BSIs were significantly more likely to have previous azole exposure and be on total parenteral nutrition than those with *C. albicans* BSIs (17.9 vs. 7.6% and 76.8 vs. 63.7%, *p* = 0.015 and 0.029, respectively). The duration of *C. parapsilosis* candidemia was relatively longer, and therefore patients often required a longer duration of antifungal treatment when compared with those of *C. albicans* candidemia, although the candidemia-attributable mortality rates were comparable. Of the *C. parapsilosis* isolates, 93.7% were susceptible to all antifungal agents, and delayed appropriate antifungal treatment was an independent factor in treatment failure. **Conclusions**: Pediatric patients with *C. parapsilosis* BSIs were more likely to have previous azole exposure and be on total parenteral nutrition, and the clinical significances included a longer duration of candidemia and patients often required a longer duration of antifungal treatment.

## 1. Introduction

Critically ill patients with long-term hospitalization in the intensive care unit (ICU), those with underlying immunocompromised status, the presence of artificial devices and the use of broad-spectrum antibiotics are vulnerable to *Candida* bloodstream infections (BSIs) [1,2,3]. A high mortality rate of 28–46% has been reported in pediatric patients with *Candida* BSIs, especially in extremely preterm neonates or those in the hematology/oncology ward [3,4,5]. In pediatric patients, multiple chronic comorbidities, breakthrough candidemia, delayed catheter removal and initial inappropriate antifungal treatment have been reported to be independently associated with treatment failure and final mortality. [6,7]. Additionally, recent studies have documented that there is an emerging trend of *Candida* isolates with antifungal resistance and the changing epidemiology from *C. albicans* to non-*albicans* candidemia is noted in the era of greater antifungal prescription [8,9,10].

*Candida parapsilosis* is an important non-*albicans Candida* species and accounts for 18–46% of all *Candida* BSIs [11,12]. *Candida parapsilosis* is well known to develop biofilm on the surfaces of artificial devices, the hands and the vulvovaginal mucosa, which results in the high likelihood of indwelling catheter-associated BSIs caused by *C. parapsilosis* [11,12,13,14]. Extremely preterm neonates need long-term central venous catheter (CVC) placement, and implanted Port-A-Cath catheters are common in pediatric hematology/oncology patients, which puts them at a higher risk of having *C. parapsilosis*-associated BSIs [15,16]. Additionally, there is an emerging resistance to azoles among *C. parapsilosis* isolates, and an increasing incidence of *C. parapsilosis* BSIs has been reported, which may pose a therapeutic challenge to clinicians [17,18]. However, the impacts of *Candida parapsilosis* BSIs on the outcomes of pediatric patients with candidemia deserve further investigation [19]. In this study, we aim to characterize *C. parapsilosis* BSIs in children and investigate the influences of therapeutic strategies on outcomes.

## 2. Patients and Methods

### 2.1. Study Design, Setting and Ethics Approval

All pediatric patients less than 18 years old who had *Candida* BSIs during hospitalization at the Linkou Chang Gung Memorial Hospital (CGMH) between January 2005 and December 2020 were enrolled and analyzed. Cases of *Candida parapsilosis* BSIs were compared with patients with *C. albicans* BSIs and other *Candida* spp. BSIs. The Linkou CGMH is the university-affiliated teaching hospital in northern Taiwan and the pediatric department of Linkou CGMH has several ICUs and a specialized hematology/oncology ward. There are a total of 24 beds and 107 beds in the pediatric ICU (PICU) and three neonatal intensive care units (NICUs) of Linkou CGMH, respectively. We continued the series of pediatric candidemia studies after the initial approval by the institutional Review Board of the CGMH (the certificate number: 202201214B0) over seven years ago. Additionally, a waiver of informed consent was also approved for anonymous data collection and the retrospective design of this study.

### 2.2. Definitions and Data Collection

In this study, we applied the following criteria to define *Candida* BSI: a patient had signs or symptoms of sepsis and ≥1 positive blood culture of *Candida* species. The *Candida* BSI episode was excluded if an unidentified *Candida* spp. was identified in the blood culture. In our institute, the clinicians usually repeat the blood cultures in cases of invasive candidiasis every 2 to 3 days until they are negative or when it is clinically indicated. The onset of *Candida* BSI was defined as the first positive blood culture of *Candida* spp. All bacterial pathogens or fungal species that were isolated while the patients were on antifungal therapy were reviewed and analyzed. If a bacterial pathogen was isolated within two days of the positive *Candida* spp. blood culture or two days after, it was considered as mixed *Candida*/bacteria BSIs. When a new infectious focus, such as an abscess, fungus ball, meningitis or end organ damage, were noted 48 h after the onset of the *Candida* BSI episode until the patient had completed the antifungal therapy or died, we considered it an infectious complication of the *Candida* BSI.

We applied the standard criteria of previous publications for the diagnosis of neonatal severe sepsis and septic shock [20,21]. When the patient completed antifungal therapy with at least two negative blood cultures from the last positive culture of the *Candida* isolate and resolution of all clinical symptoms, it was considered a new episode of *Candida* BSI if positive *Candida* spp. was isolated in the blood culture again [22]. Breakthrough candidemia was diagnosed if the new onset of *Candida* BSI occurred while this patient was still on antifungal therapy or antifungal prophylaxis [2,15].

Medical records were reviewed to determine response to antifungal therapy at two weeks after the onset of *Candida* BSIs, based on the guidelines for assessing treatment responses published by the Mycoses Study Group and the European Organization for Research and Treatment of Cancer as follows: complete response was the resolution of candidemia and clinical symptoms within 3 days; partial response was within 7 days; and progression of disease and death were considered “treatment failure” [23]. The demographic data, chronic comorbidities, hospital courses including use of antifungal agents and artificial devices and predisposing risk factors within 30 days before the onset of *Candida* BSI were also reviewed and analyzed. When the case had mortality before the resolution of signs and symptoms related to *Candida* BSIs or the patient died within 14 days after the onset of the *Candida* BSI without other explanation, it was defined as *Candida* BSI-attributable mortality [15,22]. For subsequent bacteremia following candidemia, it was defined as isolation of the bacterial pathogen between 48 h after the onset of *Candida* BSI and the time the patient completed antifungal therapy [23].

### 2.3. Microbiology and In Vitro Antifungal Susceptibility Testing

In Linkou CGMH, the BACTEC 9240 (Becton Dickinson Microbiology Systems, Franklin Lakes, NJ, USA) system is used to process all blood cultures. All *Candida* isolates from pediatric patients with *Candida* BSIs were retrieved from the central laboratory and re-identified using matrix-assisted laser desorption ionization time-of-flight mass spectrometry (MALDI-TOF, Bruker Biotype, software version 3.0, Rochester, NY, USA). The in vitro antifungal susceptibilities of isolates were evaluated according to the EUCAST-Antifungal Susceptibility Testing microdilution method [24,25]. *Candida krusei* ATCC^®^ 6258 and *Candida parapsilosis* ATCC^®^ 22019 were used as quality control strains for antifungal drug susceptibility testing.

### 2.4. Statistical Analysis

All episodes of pediatric *C. parapsilosis* BSIs during the study period were analyzed and compared with all episodes of *C. albicans* BSIs in hospitalized children from our institute. The clinical characteristics, treatment and outcomes were compared between the two groups. The demographic, clinical, outcome variables and in vitro susceptibility data were summarized using descriptive statistics. Categorical variables were compared using the χ^2^ or Fisher’s exact test, and continuous variables by the Mann-Whitney *U* test. *p*-Values < 0.05 were considered statistically significant.

The clinical significance and impacts of pediatric *Candida parapsilosis* BSIs were investigated and independent risk factors for candidemia-attributable mortality were evaluated. A univariate logistic regression was fitted for each variable to test its relationship with mortality outcomes. Variables that were clinically relevant and statistically significant (*p* < 0.1) on univariate analysis were considered for the multivariate regression model. Clinical interventions were maintained in the final model as a fixed variable. All statistical analyses were performed using IBM SPSS software (version 22.0; IBM SPSS Inc., New York, NY, USA).

## 3. Results

In the study period, there were a total of 365 episodes of *Candida* BSIs in 320 pediatric patients hospitalized in our institute, with all the *Candida* isolates re-confirmed. There were 95 episodes of *C. parapsilosis* BSIs in 88 patients, accounting for 26.0% of all pediatric *Candida* BSIs in the study period. The most common *Candida* species that caused pediatric *Candida* BSIs were *C. albicans* (n = 171, 46.8%), followed by *C. tropicalis* (n = 21, 5.8%), *C. glabrata* (n = 20, 5.5%), and *C. guilliermondii* (n = 18, 4.9%). The trends of different Candida species that caused pediatric Candida BSIs during the study period are illustrated in Figure 1. The demographics and underlying chronic comorbidities of pediatric patients with *C. parapsilosis* BSIs are summarized in Table 1. Most of the pediatric patients (89.8%) had underlying chronic comorbidities at the onset of candidemia, and 44.3% had multiple chronic comorbidities. The distributions of *C. parapsilosis* BSIs between NICU, PICU or general wards and the infectious sources were comparable with those of *C. albicans* BSIs or other *Candida* species.

At the onset of candidemia, most *C. parapsilosis* BSIs were primary bloodstream infections, but 29.5% (n = 28) were catheter-related BSIs (CRBSI), and a total of 14 (14.7%) episodes had positive *C. parapsilosis* isolates cultured from the intra-abdominal space, abscess (n = 10), pleural fluid (n = 2) and urinary source (n = 2). The *Candida* isolates were identified from more than two sterile sites in four cases of *C. parapsilosis* BSIs and were considered as disseminated candidemia. (Table 2). A total of 17 episodes of *C. parapsilosis* BSIs were breakthrough candidemia; that is, the patients were on therapeutic antifungals or antifungal prophylaxis at their disease onset.

### 3.1. Microbiological Characteristics and Clinical Features

The antifungal susceptibility results of *C. parapsilosis* isolates are summarized in Table 3. The minimum inhibitory concentrations of *C. parapsilosis* (MIC_50_) to fluconazole and voriconazole were 0.5 mg/L and 0.015 mg/L, respectively. All *C. parapsilosis* isolates were susceptible to fluconazole, voriconazole, amphotericin B and echinocandin-based antifungal regimens. There were only 1, 2 and 6 *C. parapsilosis* isolates that were resistant to itraconazole, micafungin and posaconazole, respectively. Overall, 93.7% *C. parapsilosis* isolates were susceptible to all antifungal agents.

Most of the clinical presentations, including severity of illness and percentages of severe sepsis, septic shock and disseminated candidemia, were comparable between *C. parapsilosis* BSIs and the control groups, including the *C. albicans* BSIs or other *Candida* spp. BSIs (Table 2). Most predisposing risk factors for candidemia were also comparable between the *C. parapsilosis* BSIs and the control groups. However, pediatric patients with *C. parapsilosis* BSIs were significantly more likely to have previous azole exposure and be on total parenteral nutrition (TPN) than those with *C. albicans* BSIs (17.9 vs. 7.6% and 76.8 vs. 63.7%, *p* = 0.015 and 0.029, respectively). Of note, the duration of *C. parapsilosis* BSIs was relatively longer than that of *C. albicans* BSIs (3.0 (1.0–10.0) vs. 3.0 (1.0–6.0) days, *p* = 0.068) (median [IQR] duration of candidemia).

Antifungal therapy was initiated after a median of 2 days (range, 0–7) after the onset of *Candida parapsilosis* BSIs. The initial antifungal agents and final therapeutic regimens were comparable between *C. parapsilosis* BSIs and *C. albicans* BSIs. The percentages of delayed appropriate antifungal agents and modifications of antifungal treatment were also comparable between the two groups. The median duration of antifungal treatments in patients with *C. parapsilosis* BSIs was 18.0 (IQR, 14–24) days, which was significantly longer than that of *C. albicans* BSIs (15.0 [14.0–22.0] days, *p* = 0.021). The longer duration of treatment was associated with a longer duration of candidemia and a higher rate of persistent candidemia in patients with *C. parapsilosis* BSIs. However, the candidemia-attributable mortality rates were comparable between *C. parapsilosis* BSIs, *C. albicans* BSIs and other *Candida* spp. BSIs. (Table 4). The median time between the onset of the next nosocomial infection and the previous episode of *C. parapsilosis* BSIs was 10 days (range: 4–31 days). 

### 3.2. Therapeutic Outcomes and Independent Risk Factors of Mortality

Overall, the attributable mortality rate of *C. parapsilosis* BSIs was 24.2% (23 of 95 episodes), and the in-hospital mortality rate was 33.0% (29 of 88 patients died). The therapeutic outcomes were not significantly different between different study periods, although echinocandins have been more commonly prescribed in our institute since 2010. There was no increasing trend or emergence of antifungal-resistant *C. parapsilosis* isolates during the study period in our cohort, although routine antifungal prophylaxis has been implemented for very low birth weight (VLBW, birth weight ≤ 1500 g) neonates in our NICUs since 2015. Additionally, the therapeutic responses were comparable between patients with *C. parapsilosis* BSIs and those with *C. albicans* BSIs and other Candida spp. BSIs. 

The independent risk factors of candidemia-attributable mortality were investigated in this study (Table 5). Neonates had relatively higher rates of sepsis-attributable mortality and in-hospital mortality than children with *C. parapsilosis* BSIs (Table 1). Delayed catheter removal (>72 h after onset of *Candida* BSIs), subsequent bacteremia after *Candida* BSIs, breakthrough candidemia, more chronic comorbidities and septic shock at onset were enrolled into the multivariable regression model for their significant association with an increased risk of candidemia-attributable mortality. After multivariate logistic regression analyses, the independent risk factors for candidemia attributable mortality in pediatric patients with *C. parapsilosis* BSIs were septic shock at onset (OR, 5.75; 95% CI: 2.08–10.14, *p* < 0.001), breakthrough candidemia (OR, 3.58; 95% CI: 1.93–8.87, *p* = 0.002), and delayed catheter removal (OR, 2.86; 95% CI: 1.16–7.05, *p* = 0.022).

## 4. Discussion

The trend of increasing non-*albicans* candidemia was noted in our institute and has also been documented in other countries in the literature [11,12,26]. Although *C. parapsilosis* BSIs in children were not significantly associated with worse outcomes, we found that pediatric patients with *C. parapsilosis* BSIs were more likely to have a longer duration of candidemia, slower responses to antifungal treatment and require a longer duration of antifungal treatment when compared with those with *C. albicans* BSIs. In children with *Candida parapsilosis* BSIs, the antifungal resistance rate was only 14% of all episodes [7], and only a few episodes had delayed initial appropriate treatment. We were unable to identify any independent risk factors for *C. parapsilosis* BSIs because most characteristics were comparable with the controls. However, *C. parapsilosis* candidemia deserves more concern since the high percentage of NICU patients with CVCs could potentially cause its increasing prevalence in the future.

The proportion of *C. parapsilosis* candidemia ranged between 14% and 34% of all *Candida* BSIs in both children and adults [11,12,13,14,18,26]. In recent years, non-*albicans* candidemia has outnumbered *C. albicans* candidemia, especially among critically ill patients, neonates with long-term CVC placement, those on TPN and those with antifungal prophylaxis [1,4,5,26,27]. Previous studies have found the use of CVC or other artificial devices, such as urinary catheters or mechanical ventilators, to be associated with *C. parapsilosis* candidemia [19,28,29,30], while other studies also found patients receiving TPN were more likely to have *C. parapsilosis* CRBSI [11,31]. Our results were consistent with previous studies, although we could not find significant associations between CVC use and *C. parapsilosis* BSIs, which may be explained by the presence of CVC in almost all pediatric patients at the onset of candidemia. Additionally, a history of prior antifungal therapy was found to be associated with *C. parapsilosis* candidemia [30,32], which may be due to the effects of antifungal prophylaxis on the selection of non-*albicans* species, especially after echinocandin administration [32,33]. 

*C. parapsilosis* is known to exhibit a strong biofilm forming capability on the surfaces of a CVC or other artificial devices, which accounts for its higher prevalence among neonates with low birth weight, ICU patients with long term use of CVC and those who are immunocompromised [11,12,13,14,34]. Therefore, the most effective control strategy to avoid persistent candidemia is early catheter removal [7,16,34], although sometimes it is not applicable. Based on our results, we found *C*. *parapsilosis* BSIs had a significantly longer duration of candidemia, and delay in catheter removal was independently associated with final adverse outcomes. Previous studies have also found that patients with recurrent or breakthrough candidemia were more likely to have persistent candidemia and final treatment failure [15,35,36], which are supposed to be associated with the placement of CVCs. Given the inevitable requirement of a CVC in VLBW neonates, cases of *C. parapsilosis* BSIs deserve greater attention and further research for species-specific strategies is warranted.

Most studies found that patients with *C. parapsilosis* BSIs have comparable outcomes when compared with those with *C. albicans* BSIs or other non-*parapsilosis* candidemia [14,28,29,34], while others even found that the mortality rate was lower than that of *C. albicans* BSIs [37,38]. Overall, the different therapeutic strategies, study cohorts and different underlying demographics may account for the controversial results [11,12,13,14,24,25,26,27,28,29,30,31,32,33,34,35,36,37,38]. The candidemia-attributable mortality rate in our cohort was relatively higher than that of previous studies [4,5,30,31,32,33,34], especially in extremely preterm neonates. Although a new antifungal agent, echinocandin or caspofungin, has been launched in our institute since nearly a decade ago, this anti-biofilm agent did not significantly improve outcomes. We suspected that high percentages of chronic comorbidities, especially multiple chronic comorbidities, may account for the poor outcomes in our cohort. Although only a few *C. parapsilosis* isolates were antifungal-resistant, there has been an emergence of antifungal resistant *C. parapsilosis* isolates worldwide [39,40]. Therefore, *C. parapsilosis* BSIs deserves greater attention and specific therapeutic strategies for cases with a high risk of treatment failure should be investigated to optimize outcomes in the future.

There were some limitations in this study. The retrospective nature and single-center data limit the conclusion to being generalizable and applicable to other institutes or countries. Due to the long study period, there was inevitably some missing data and lost *Candida parapsilosis* isolates, which would make the incidence rate less reliable. The therapeutic strategies may have changed during the long study period, although the outcomes of *C. parapsilosis* BSIs were similar in the past two decades. Additionally, the pediatric cases of *Candida* BSIs in our institute did not have a regular schedule of follow-up blood cultures and the therapeutic strategies depended on the decisions of the attending physicians. Therefore, a further large-scale prospective study is required to address the risk factors and real impacts of *C. parapsilosis* on the outcomes of pediatric patients with *Candida* BSIs.

In conclusion, *C. parapsilosis* isolates are the most common non-*albicans* candidemia in children, and the use of CVC and other artificial devices, as well as the current antifungal prophylaxis policy, may predispose children to have *C. parapsilosis* BSIs. The mortality rate of pediatric patients with *Candida* BSIs in children was high, and the chance of increasing azole resistance in Candida isolates deserves more concern. Since early catheter removal is especially important to avoid persistent candidemia and significantly affect final outcomes, clinicians should not treat *C. parapsilosis* BSIs with catheters in situ. In cases of multiple chronic comorbidities or septic shock, clinicians should consider more aggressive therapies.

## Figures and Tables

**Figure 1 microorganisms-11-01149-f001:**
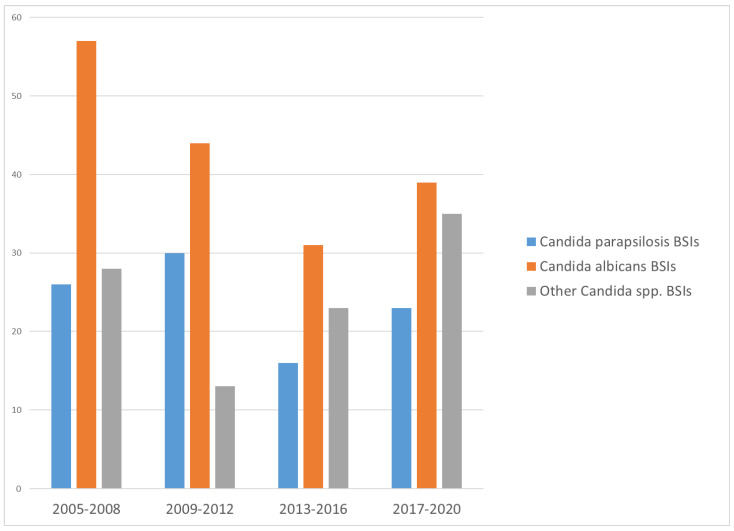
The case distributions of different *Candida* species that caused pediatric *Candida* bloodstream infections (BSIs) in the study period, 2005–2020, in Chang Gung Memorial Hospital.

**Table 1 microorganisms-11-01149-t001:** Demographic and clinical data of pediatric patients with *Candida parapsilosis* bloodstream infections in CGMH, 2005–2020.

Patients Characteristics (Total n = 88)	Neonates (Age < 3 months) (n = 29)	Pediatric Patients (n = 59)	*p*-Value
Patients demographics			
Birth body weight (g), median (IQR)	1220.0 (740.0–2150.0)	-	-
Gestational age (weeks), median (IQR)	29.0 (26.5–35.5)	-	-
Age (years old), median (IQR)	-	4.7 (1.8–13.5)	-
Gender (male/female)	18 (62.1)/11 (37.9)	33 (55.9)/26 (44.1)	0.625
NSD/Cesarean section	13 (44.8)/16 (55.2)	-	-
Inborn/outborn	23 (79.3)/6 (20.7)	-	-
5 min Apgar score ≤ 7, n (%)	11 (37.9)	-	-
Perinatal asphyxia, n (%)	2 (6.9)	-	-
Respiratory distress syndrome (≥Gr II), n (%)	16 (55.2)	-	-
Intraventricular hemorrhage (≥Stage II), n (%)	6 (20.7)	-	-
Day of life at onset of candidemia (day), median (IQR)	28.0 (13.0–68.0)	-	-
Duration of hospitalization before onset of candidemia (day), median (IQR)	-	32.5 (14.0–51.0)	-
Underlying chronic comorbidities, n (%)			-
Neurological sequelae	8 (27.6)	23 (39.0)	0.213
Bronchopulmonary dysplasia or chronic lung disease	14 (48.3)	10 (16.9)	0.012
Complicated cardiovascular diseases	2 (6.9)	3 (5.1)	0.726
Symptomatic patent ductus arteriosus	6 (20.7)	0 (0)	-
Gastrointestinal sequelae	4 (13.8)	20 (33.9)	0.041
Renal disorders	3 (10.3)	6 (10.1)	0.976
Hematological/Oncology	0 (0)	10 (16.9)	-
Immunodeficiency	0 (0)	2 (3.4)	-
Autoimmune disorders	0 (0)	3 (5.1)	-
Congenital anomalies	4 (13.8)	10 (16.9)	0.408
Presence of any chronic comorbidities	23 (79.3)	56 (94.9)	0.075
Presence of more than one comorbidities	11 (37.9)	28 (47.5)	-
Case years			-
2005–2008	8 (24.2)	18 (29.0)	
2009–2012	11 (33.3)	19 (30.6)	-
2013–2016	6 (18.2)	10 (16.1)	-
2017–2020	8 (24.2)	15 (24.2)	-
Candidemia-attributable mortality	10/33 (30.3)	13/62 (21.0)	0.185
Overall final in-hospital mortality	13 (44.8)	16 (27.1)	0.147

BSI: bloodstream infection; IQR: interquartile range; NSD: normal spontaneous delivery.

**Table 2 microorganisms-11-01149-t002:** Comparisons of *Candida parapsilosis* bloodstream infections (BSIs) and *Candida albicans* and other *Candida* spp. BSIs in CGMH, 2005–2020.

	*Candida parapsilosis* BSIs (n = 95)	*Candida albicans* BSIs (n = 171)	Other *Candida* spp. BSIs (n = 99)	*p*-Value *
Neonatal episodes	33 (34.7)	53 (31.0)	29 (29.3)	0.744
Pediatric episodes	62 (65.3)	118 (69.0)	70 (70.7)	
Source of candidemia **				0.526, 0.028
Primary bloodstream infection (BSI)	53 (55.8)	103 (60.2)	70 (70.7)	0.517, 0.037
Catheter-related BSI	28 (29.5)	38 (22.2)	12 (12.1)	0.236, 0.004
Abdominal	10 (10.5)	15 (8.8)	12 (12.1)	
Urological	2 (2.1)	6 (3.5)	1 (1.0)	
Pulmonary	2 (2.1)	6 (3.5)	1 (1.0)	
Meningitis	0 (0)	3 (1.8)	3 (3.1)	
Clinical presentation				
Sepsis	78 (82.1)	140 (81.9)	83 (83.8)	0.918, 0.884
Severe sepsis	44 (46.3)	64 (37.4)	41 (41.4)	0.457, 0.866
Septic shock	27 (27.3)	50 (29.2)	28 (28.3)	0.982, 0.874
Progressive and deteriorated ^¶^	22 (23.2)	38 (22.2)	12 (12.1)	0.873, 0.082
Disseminated candidiasis ^$^	4 (4.2)	10 (5.8)	3 (3.0)	0.461, 0.555
Duration of candidemia (days), median (range)	3.0 (1.0-32.0)	3.0 (1.0–20.0)	3.0 (1.0–30.0)	0.068, 0.830
Breakthrough candidemia	17 (17.9)	12 (7.0)	18 (18.2)	0.012, 0.846
Predisposing risk factors ^#^				
Receipt of systemic antibiotics ^&^	88 (92.6)	158 (92.4)	94 (94.9)	1.000, 0.562
Previous azole exposure ^&^	17 (17.9)	13 (7.6)	10 (10.1)	0.015, 0.147
Previous bacteremia ^&^	45 (47.3)	80 (46.8)	59 (59.6)	1.000, 0.113
Presence of CVC	94 (98.9)	162 (94.7)	98 (99.0)	0.102, 1.000
Receipt of parenteral nutrition	73 (76.8)	109 (63.7)	68 (68.7)	0.029, 0.259
Receipt of immunosuppressants	10 (10.5)	33 (19.3)	27 (27.3)	0.081, 0.003
Artificial device other than CVC	48 (50.5)	75 (43.9)	57 (57.6)	0.307, 0.387
Prior surgery ^&^	26 (27.4)	52 (30.4)	33 (33.3)	0.578, 0.436
Neutropenia (ANC < 0.5 × 10^3^/μL)	23 (24.2)	35 (20.5)	29 (29.3)	0.536, 0.517

All data are expressed as case number (%), unless otherwise stated. * *p*-Values are comparisons between patients with *C. parapsilosis* BSIs and *C. albicans* BSIs, and those with *C. parapsilosis* BSIs and other *Candida* spp. BSIs. ** Source of candidemia was defined as the first sterile site to have a positive culture for the *Candida* species in the episode. ^¶^ Defined as candidemia episodes with more disseminated candidiasis and/or progressive multi-organ failure even after effective antifungal agents. ^#^ Indicated the presence of an underlying condition or risk factor at the onset of *Candida* BSI, and most episodes occurred in patients with >1 underlying condition or risk factor. ^&^ Within one month prior to the onset of invasive candidemia. ^$^ Indicated positive *Candida* isolates recovered from more than two sterile sites in addition to primary bloodstream infection. CVC: central venous catheter; ANC: absolute neutrophil count.

**Table 3 microorganisms-11-01149-t003:** Distributions of *Candida parapsilosis* isolates from pediatric patients of CGMH according to minimum inhibitory concentration values calculated for different antifungals.

Pathogens/Antifungals	No. of Isolates with MIC (mg/L) of *Candida parapsilosis* (n = 95 Tested)											MIC (mg/L)		
	0.008	0.015	0.03	0.06	0.12	0.25	0.5	1.0	2.0	4.0	≥8.0	GM	MIC_50_	MIC_90_
Fluconazole					2	7	40	36	10			0.569	0.5	2.0
Itraconazole		2	19	44	29	1						0.063	0.06	0.12
Voriconazole	19	44	26	6								0.017	0.015	0.03
Posaconazole	2	11	47	29	6							0.037	0.03	0.06
5-Flucytosine				17	43	29	4	2				0.148	0.12	0.25
Amphotericin B						8	55	32				0.598	0.5	1.0
Micafungin			1				17	53	22	2		1.015	1.0	2.0
Caspofungin			1		1	6	56	27	4			0.592	0.5	1.0
Anidulafungin			1	1		1	16	60	16	1		0.928	1.0	2.0

MIC: minimum inhibitory concentration. MIC_50_ and MIC_90_: MIC required to inhibit 50% and 90% of the isolates, respectively. GM: geometric mean. The MIC cutoff values of *C. parapsilosis* with antifungal resistance to micafungin, itraconazole and posaconazole are ≥4.0, ≥0.25 and ≥0.12 mg/L, respectively.

**Table 4 microorganisms-11-01149-t004:** Therapeutic strategies and outcome comparisons of *Candida parapsilosis* bloodstream infections (BSIs) and *Candida albicans* and other *Candida* spp. BSIs in CGMH, 2005–2020.

Variable	*Candida parapsilosis* BSIs (n = 95)	*Candida albicans* BSIs (n = 171)	Other *Candida* spp. BSIs (n = 99)	*p*-Value *
Final treatment regimens				0.134, 0.011
Fluconazole/Voriconazole	39 (41.1)	70 (40.9)	28 (28.3)	0.989, 0.050
Amphotericin B	30 (31.6)	50 (29.2)	26 (26.3)	0.780, 0.545
Echinocandin-based regimen	20 (21.1)	40 (23.4)	43 (43.4)	0.847, 0.009
Combined antifungal treatment	5 (5.3)	2 (1.2)	1 (1.0)	
No treatment	1 (1.1)	9 (5.3)	1 (1.0)	
Modification of antifungal agents	35 (36.8)	64 (37.4)	58 (58.6)	0.790, 0.003
Duration of antifungal treatment (d), median (IQR)	18.0 (14.0–24.0)	15.0 (14.0–22.0)	18.0 (14.0–24.0)	0.021, 0.568
Early removal of a central venous catheter **	29 (30.5)	67 (39.2)	32 (32.3)	0.183, 0.877
Appropriate antifungal treatment within 48 h	52 (54.7)	94 (55.0)	59 (59.6)	0.794, 0.559
Treatment outcomes				
Responsiveness after effective antifungals ^&^				0.248, 0.123
Within 72 h	33 (34.7)	81 (47.4)	22 (22.2)	
4–7 days	17 (17.9)	27 (15.8)	28 (28.3)	
More than 7 days	17 (17.9)	23 (13.5)	23 (23.2)	
Treatment failure	28 (29.5)	40 (23.4)	26 (26.3)	
Subsequent bacteremia	27 (28.4)	36 (21.1)	25 (25.3)	0.180, 0.631
Candidemia-attributable mortality	23 (24.2)	40 (23.4)	24 (24.2)	0.881, 1.000

* *p*-Values are comparisons between patients with *C. parapsilosis* BSIs and *C. albicans* BSIs, and those with *C. parapsilosis* BSIs and other *Candida* spp. BSIs. ** Within 3 days after the onset of candidemia. ^&^ Responsiveness to antifungal agents was defined according to the consensus criteria of the Mycoses Study Group and the European Organization for Research and Treatment of Cancer [23].

**Table 5 microorganisms-11-01149-t005:** Univariate and multivariate logistic regression analysis for independent risk factors of candidemia-attributable mortality in pediatric patients with *Candida parapsilosis* bloodstream infection.

Variables	Univariate Analyses			Multivariate Regression Analysis		
	Odds Ratio	95% CI	*p*-Value ^#^	Odds Ratio	95% CI	*p*-Value ^#^
Patients’ age						
Neonates (<3 months old)	2.05	1.14–3.67	0.016			
Pediatric patients	1	(reference)				
Underlying chronic comorbidities						
No	1	(reference)		1	(reference)	
One	1.03	0.44–3.08	0.433	1.04	0.54–3.34	0.219
More than one chronic comorbidity	2.34	0.74–7.37	0.090	2.81	0.71–6.64	0.118
Septic shock at onset	10.44	4.76–25.36	<0.001	5.75	2.08–10.14	<0.001
Delayed CVC removal (>72 h)	4.60	2.49–8.78	<0.001	2.86	1.16–7.05	0.022
Subsequent bacteremia	1.73	0.92–3.24	0.087	1.55	0.45–3.61	0.644
Breakthrough candidemia	4.35	2.63–9.17	<0.001	3.58	1.93–8.87	0.002
Delayed effective antifungal agents (>48 h)	1.57	0.85–2.85	0.150			
Pathogens						
*Candida albicans*	1	(reference)				
*Candida parapsilosis*	0.956	0.53–1.72	0.880			

CI: confidence interval; CVC: central venous catheter. ^#^ Hosmer-Lemeshow *p* = 0.671 for candidemia attributable mortality.

## Data Availability

The datasets used/or analyzed during the current study are available from the corresponding author on reasonable request.

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
