# Peer review of "Invasive Candida parapsilosis Bloodstream Infections in Children: The Antifungal Susceptibility, Clinical Characteristics and Impacts on Outcomes"

_microorganisms, 2023, doi:10.3390/microorganisms11051149_

Round 1

Reviewer 1 Report

The authors report the cases of C. parapsilosis Blood Stream Infection (BSI) in the pediatric population of a hospital in Taiwan between 2005 and 2020. The impact of the antifungal sensitivity of the isolates, as well as the clinical characteristics of the patients, in relation to the outcome is analyzed.

There is evidence that in the pediatric population C. parapsilosis is the most frequent species in BSIs produced by non-albicans Candida species. They conclude that compared to pediatric patients with C. albicans BSIs, those with C. parapsilosis isolates were more frequent among patients who had received prior treatment with azoles and on total parenteral nutrition, and they experienced a longer duration of candidemia, often requiring longer antifungal treatments.

INTRODUCTION

- The authors state “Candida parapsilosis is the most common non-albicans Candida species and account for 18-46% of all Candida BSIs [11,12]”. This statement does not conform to reality, since the second position among the isolates does not correspond to all geographic locations, nor to the clinical characteristics of the populations studied (adults, pediatric patients, catheter-associated infections...) as described in reference 12.

- “However, Candida parapsilosis BSIs hasn’t been well investigated in pediatric patients with candidemia [19].” Do you mean that research performed up to date is not appropriate? PubMed search for “candida parapsilosis AND paediatric” retrieves 349 results… Please rewrite your sentence.

PATIENTS AND METHODS

The sections contain descriptions that do not correspond to the titles. It is suggested that the sections in this section be rearranged and renamed, in addition to revising some aspects detailed below.

2.1. Study design, setting and ethics approval.

This section should be rewritten as it contains unconnected sentences, a description of the Hospital units with non-concordant data. It includes methodology of the microbiology laboratory that is not related to the title of the section.

Describe the meaning of the acronyms for the different sections of the pediatric intensive care ward (later in the manuscript, page 4, you mention NICU, PICU, general wards…)

2.2. Definitions and data collection

The wording is confusing, please rewrite. Definitions are not clear, names of microorganisms should be written with initial capital letters and italicized characters. Microbiology data (antifungal sensitivity tests) Microbiology data (antifungal sensitivity tests) should go in a separate section.

2.3. Statistical analysis

What is the criterion for setting the significance level of the univariate analysis at P<0.1 for inclusion in the multivariate analysis? In fact, the significant factors in the latter analysis are those that were already significant (P<0.05) in the univariate analysis.

RESULTS

-          According to Table2, total Candida BSIs episodes were not 356, as mentioned in the first line of the results section.

-          In general, the description of the most relevant data is confusing.

-          Figure 1. The graph does not show the trend, but the frequency of isolates of different Candida species in BSIs throughout the study. The trend could be deduced from it, but these are absolute values.

-          Table 1. It shows not only patients' demographics but also clinical data that should be divided into categories. The main categories in the first column should be left-justified and show at a lower level the elements that correspond to each category.

-          Table 2 requires the same reorganization of the characteristics in the first column. There are discrepant data between Table 1 and 2: for example, the number of episodes of C. parapsilosis BSIs in the neonatal group (33 or 32?) and pediatric patients (62 or 63?). The p values column presents unaligned data.

Describe the significance of ANC, space out footnote explanations.

-          Since clinical and demographic data of BSIs by different Candida species are compared, it would be useful to know the data shown in Table 1 for patients with isolates of C. albicans and other species, at least as supplementary material.

3.1. Microbiological characteristics and clinical features

- MIC50 for fluconazole does not match between the text and Table 3.

- Table 3 description should read: Distribution of C. parapsilosis isolates according to the MIC values calculated for the different antifungals.

- Table 4. Same indications as in Tables 1 and 2 for the first column. The p values column presents non-aligned data.

3.2. Therapeutic outcomes and independent risk factor of mortality

- The inclusion of factors with p<0.1 in the multivariate analysis does not seem to make any difference with the univariate analysis, since significant factors (p<0.05) are the same in both studies.

DISCUSSION

The authors confront their results with those of the literature and highlight the limitations of the study.

The conclusions are derived from their results and are in line with other similar studies.

ENGLISH LANGUAGE AND STYLE

The wording in English needs to be revised, especially some paragraphs highlighted in the attached document.

Some of the comments previously mentioned are also marked in this document.

Some phrases are difficult to understand due to missing or misspelled words.

Please, revise the correct spelling of species names, and use italicized characters style every time you refer to Candida and its species.

Author Response

Dear reviewer:

    I appreciate your review and comments. Please see the attached file, thank you.

Best regard,

Tsai Ming Horng

Reviewer 2 Report

Clarify if a patient had several positive cultures which one or which were taken into account for this study?

Dis you get a blood culture sample that had 2 species of Candida at the same time?

Please include in the information of suscpetibility tests which minimum inhibitory concentration values were taken as suscpetible, intermediate, and resistant,

Give some information about the correlation between susceptibility test results and treatment effect. That is, the authors are reporting strains resistant to itraconazole, posaconazole, and micafungin. Was the MIC results really predictive?

Author Response

(The authors gave the same response as above.)

Round 2

Reviewer 1 Report

The authors have addressed most of this reviewer's comments satisfactorily; however, some aspects remain to be clarified, as well as some details that could contribute to improve the final presentation of the paper. I’m sending the manuscript with some areas outlined to easy the revision.

PATIENTS AND METHODS

1. Regarding the new Microbiology section (2.3), for a more logical presentation, the second sentence “In Linkou CGMH, the BACTECall blood cultures”, would be more adequately inserted before “All Candida isolates…”, as it was in your first version.

2. The sentence regarding “ P value” is not correct.

RESULTS

3. You wrote “Would you please point out directly which part of the description of the relevant data will cause confusing if there is next round revision. I appreciate your help, thank you”.

Now that tables show categorized data, so it is easier to locate the results described. However, for easier reading, some numerical data already in the tables could be removed (for example: some IQR, choose between numbers and percentage in some cases...).

4. Figure 1. Please, remove the heading “The trends….” inserted in the graphic.

5. Since Table 1 includes demographic and clinical data, the title should read “Demographic and clinical data of pediatric patients…”.

Here you may find the argument for this requirement: Demographic data refers to information related to the characteristics of the population being studied, such as age, gender, ethnicity, education, socio-economic status, occupation, and place of residence. These data are used to characterize the study population and may be important for understanding the prevalence and distribution of a disease or condition in different demographic groups. Additionally, demographic data may also be useful for identifying population groups with greater healthcare needs and designing appropriate prevention and treatment strategies. Clinical data typically refers to information related to a patient's medical history, such as symptoms, diagnoses, medications, and laboratory results, and is not usually considered part of demographic data. While clinical data may provide important insights into a patient's health status and may be relevant for specific health studies, it is not typically included under the umbrella of demographic data. That being said, some studies may include both demographic and clinical data, particularly when investigating the relationship between demographic characteristics and health outcomes. However, it is important to clearly distinguish between demographic and clinical data in order to accurately analyze and interpret the results of a study.

6. Tables 1 and 2: the number of episodes of C. parapsilosis BSIs in the neonatal group (33 or 32?) and pediatric patients (62 or 63?) differ in both tables. The p values column presents unaligned data.

I do not agree with your comment “The table 1 is for patients’ characteristics, and table 2 is for episodes of Candida BSIs’ characteristics. There were a total of 32 episodes in 29 neonates and 63 episodes in 59 pediatric patients”. These are the numbers in Table 2, but in Table 1, the sum of “case years” is different (33 and 62 respectively), please review numbers.

Regarding your answer “For the p values in Table 2 which you marked in the PDF file and think it unaligned data, it actually represents the comparisons of the item: Source of candidemia, the overall comparisons between the groups”, I have an additional comment: if so, with reference to the “Non-aligned p values” for “Source of candidemia”, I do not understand what the compared precise data for each group are. Please, specify data and/or statistical test applied.

Moreover, in order to focus attention on statistically significant factors, it would be helpful to outline (i.e. bold characters) p-values <0.05 in tables.

7. For Table 4, similarly to Table 2, with reference to the “Non-aligned p values”, I do not understand what the compared precise data of each group in main categories are (Final treatment regimens, Responsiveness after effective antifungals). Please, specify data and/or statistical test applied.

8. Moreover, in Table 4, is there any reason for showing no p-value for amphotericin B treatment comparison?

Author Response

Dear reviewer:

      Please see the attachment. I appreciate your review and comments, thank you.

Best regard,

Tsai Ming Horng
